# Improvements in Probabilistic Strategies and Their Application to Turbomachinery

Andriy Prots [ID], Matthias Voigt * and Ronald Mailach

Chair of Turbomachinery and Flight Propulsion, Dresden University of Technology, 01069 Dresden, Germany; andriy.prots@tu-dresden.de (A.P.); ronald.mailach@tu-dresden.de (R.M.)
* Correspondence: matthias.voigt@tu-dresden.de

**Abstract:** This paper discusses various strategies for probabilistic analysis, with a focus on typical engineering applications. The emphasis is on sampling methods and sensitivity analysis. A new sampling method, Latinized particle sampling, is introduced and compared to existing sampling methods. While it can increase the quality of surrogate models, an optimized Latin hypercube sampling is mostly preferable as it shows slightly better results. In sensitivity analysis, the difficulty lies in correlated input variables, which are typical in engineering applications. First, the Sobol indices and the Shapley values are explained using an intuitive example. Then, the modified coefficient of importance is introduced as a new sensitivity measure, which can be used to reliably identify input variables without functional influence. Finally, these results are applied to a turbomachinery test case. In this case, the flow field of a compressor row is investigated, where the blades are subjected to geometric variability. The profile parameters used to describe the geometric variability are correlated. It is shown that the variability of the maximum camber and the thickness of the leading edge have a decisive influence on the variability of the isentropic efficiency.

**Keywords:** probabilistic; turbomachinery; sampling; sensitivity analysis

## 1. Introduction

Probabilistic studies have been gaining increased importance in engineering applications, especially in the field of turbomachinery. They allow one to consider the variability of components so that their impact on result values can be quantified. Different methods can be used like sensitivity analysis, which can help to understand how changes in input parameters can affect the performance of turbomachinery components like compressors and turbines. For example, Lange et al. [1] investigated the effect of manufacturing variability on the performance of a high-pressure compressor. The blades were measured and parameterized to describe their geometric variability. This allowed them to assess the impact on performance parameters such as pressure ratio and efficiency. In the following years, this approach has been extended and applied to other components like turbines (Voigt et al. [2] and Högner et al. [3,4]). The results from such an analysis can be fed back into the manufacturing process to implement improvements that reduce the impact of geometric variability. Other examples of uncertainty quantification and sensitivity analysis are Seshadri et al. [5], Lavagnoli et al. [6], Ghisu and Shahpar [7], and Fiedler et al. [8]. Furthermore, these results can be used within a (robust) optimization (such as in Verstraete et al. [9], Padulo et al. [10], Seshadri et al. [11], Dow and Wang [12], Kamenik et al. [13], and Dittmann et al. [14]).

One main problem in probabilistic studies is the increased numerical effort. In contrast to the deterministic approach, numerical experiments such as CFD simulations have to be carried out many times. To overcome this issue, surrogate models can be used. They can be evaluated much faster so that probabilistic studies are accelerated. However, a high-quality surrogate model is desirable. For this, a suitable sampling method must be selected. In this paper, different sampling methods are discussed with respect to the surrogate model quality.

Another issue arises for sensitivity analysis. For most methods, uncorrelated input variables are required. However, correlated input variables can be present in engineering applications. Therefore, suitable methods are needed to account for correlated input variables. In this paper, suitable methods for sensitivity analyses of engineering applications are selected.

This paper is structured as follows. First, the theoretical foundations are given in Section 2. Then, different sampling methods are discussed in Section 3, with the main focus being on surrogate model quality. Section 4 discusses different methods of sensitivity analysis, especially in terms of their application to problems with correlated input variables. In Section 5, a sensitivity analysis of a turbomachinery test case is performed where the conclusions of the previous two sections are applied. This paper is closed with a short summary in Section 6.

## 2. Theoretical Foundation

In this section, a brief introduction into probabilistic methods is given. Further details can be found in the standard works of statistics like in Montgomery and Runger [15]. Section 2.1 presents the general steps of a probabilistic study. Section 2.2 presents the different types of surrogate models that are used in this paper. Section 2.3 discusses the different metrics of the surrogate model quality.

### 2.1. Monte Carlo Simulation and Probabilistic Studies

Probabilistic methods are used to analyze systems, which are subject to variability or uncertainty. For example, compressor blades are subject to manufacturing variability and wear, which changes their geometric shape. This, in turn, has an impact on the flow field and thus performance parameters like pressure ratio and efficiency. Probabilistic methods can be used to quantify the impact of the variability or uncertainty of input variables.

Monte Carlo simulation (MCS) is a well-established method in this context. First, the problem must be described mathematically and statistically. For this, the variability of the examined objects must be captured by analyzing the actual components (Figure 1a). In the context of compressor blades, optical or tactile measurements are usually performed. In the next step, the captured data and their variability are described by parameters (Figure 1b). To reduce the complexity of the subsequent steps, it is desirable to have as few parameters as possible. Furthermore, these parameters should be interpretable by engineers such that conclusions can be drawn from the statistical results. By analyzing the entire data set, the parameters are statistically described. This includes both the marginal distributions of the parameters and the correlations between them (Figure 1c).

Now, the numerical and statistical evaluation of the problem is performed. Based on the statistical description of the input variables, a sample is generated (Figure 1d). For the sample size, a compromise must be found between accuracy (large sample size) and numerical effort (low sample size). Furthermore, for statistical analysis, the marginal distributions should be represented well. For each realization of the sample (represented by a dot), the deterministic model is evaluated to capture the system response for the given values of the input parameters. Examples of deterministic models are CFD calculations (Figure 1e) or the estimation of stresses and strains obtained using the finite element method (FEM). For each realization, the result value of interest, such as the efficiency, is extracted. In the final step, the entire sample is evaluated, yielding statistical measures like the mean and standard deviation of the result value of interest (Figure 1f).

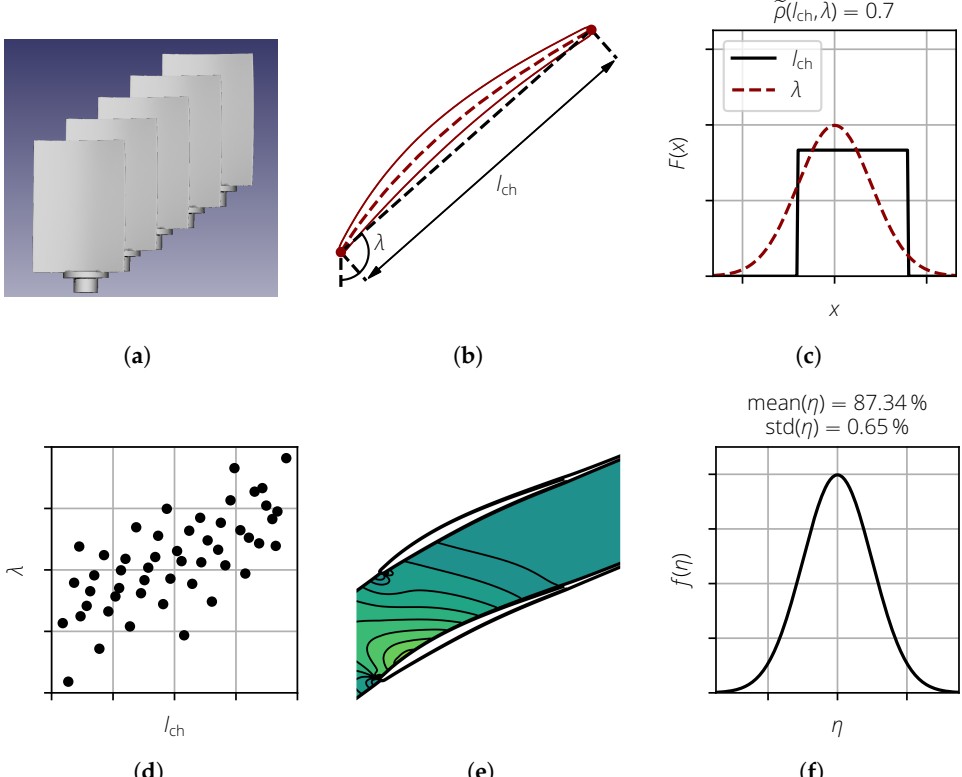

**Figure 1.** Steps of a Monte Carlo Simulation on an example analysis of compressor blades. (**a**) Data acquisition; (**b**) parameterization; (**c**) statistical description; (**d**) sampling; (**e**) deterministic calculations; and (**f**) statistical evaluation.

The results of an MCS can now be used for further investigations. For example, using its results, surrogate models can now be created, which can provide an accurate approximation of the deterministic model, but they can also be evaluated much faster. This enables further statistical evaluations, like sensitivity analyses, to identify the input variables whose variance has the greatest influence on the variability of the result variables. Another example is optimization. In classical optimization, one or more target variables are improved. In a robust optimization, an additional goal is to ensure that these target variables change only slightly when input variables change.

### 2.2. Surrogate Models

In probabilistic studies, the underlying deterministic models are usually complex, and their evaluation is both time and computationally intensive. An alternative to this are surrogate models, which can be evaluated much faster and can significantly reduce the computational time.

With surrogate models, the relationship $y = f(x)$ between the (multidimensional) input variable $x$ and result value $y$ of the deterministic model is represented by a simplified mathematical formulation. Typically, a surrogate model cannot exactly reproduce the simulation model, and a surrogate model error $\varepsilon = y - \widetilde{y}$ remains, where $y$ is the actual response of the deterministic model and $\widetilde{y}$ is the response predicted by the surrogate model. In the following, the polynomial surrogate model (Section 2.2.1) and the Gaussian process (GP, Section 2.2.2) are discussed as they are used extensively in this work.

2.2.1. Polynomial Surrogate Models

A polynomial surrogate model is a linear model of the form

$$\widetilde{f}(x_1, x_2, \ldots, x_{n_d}) = \widetilde{f}(x) = \sum_{j=1}^{n_k} c_j \phi_j(x), \tag{1}$$

where the $n_k$ basis functions $\phi_j(x)$ are polynomials with respect to the input variables. Equation (1) can be written in the following matrix form:

$$\widetilde{f}(x) = bc,\tag{2}$$

with $b = \begin{pmatrix} \phi_1(x) & \phi_2(x) & \dots & \phi_{n_k}(x) \end{pmatrix} \in \mathbb{R}^{1 \times n_k}$ and $c = \begin{pmatrix} c_1 & c_2 & \dots & c_{n_k} \end{pmatrix}^T \in \mathbb{R}^{n_k \times 1}$.

The most common way to determine the coefficient $c$ is via the least squares method, where the residual sum of squares $SS_{\text{res}}$ is minimized as follows:

$$SS_{\text{res}} = \sum_{i=1}^{n_{\text{sim}}} \varepsilon_i^2 = \sum_{i=1}^{n_{\text{sim}}} (y_i - \widetilde{y}_i)^2.\tag{3}$$

The least squares solution for $c$ is given by

$$c = (B^T B)^{-1} B^T y.\tag{4}$$

### 2.2.2. Gaussian Process

The GP is a Bayesian approach to regression that is becoming increasingly popular in the machine learning community. It is defined as a set of infinitely many random variables, where each finite subset has a multidimensional normal distribution (Rasmussen and Williams [16]). In this section, the GP is briefly discussed. A detailed mathematical description is given by Rasmussen and Williams [16].

The GP is defined by a mean function $m(x)$ and a covariance function $k(x, x')$ (also called kernel), with

$$m(x) = \mathbb{E}[f(x)],\tag{5}$$
$$k(x, x') = \mathbb{E}\big[(f(x) - m(x))(f(x') - m(x'))\big].\tag{6}$$

The covariance function provides a mathematical formulation for the covariance between two distinct points $x$ and $x'$. In this paper, the following Gaussian kernel is used:

$$k(x, x') = \sigma_f^2 \exp\left(-\frac{1}{2l_f^2}\| x - x'\|_2^2\right).\tag{7}$$

Here, $\sigma_f^2$ is the signal variance, $l_f$ is the (multidimensional) length scale, and $\|x - x'\|_2$ is the Euclidean distance between $x$ and $x'$. Additionally, a noise variance $\sigma_y$ is introduced to consider the noise in the data. Further covariance functions are given by Rasmussen and Williams [16].

The training of the model parameters $\theta = \{l_f, \sigma_f, \sigma_y\}$ is performed by using the maximum likelihood method. The model parameters are chosen to maximize the log-marginal likelihood as follows:

$$\log(p(y|X, \theta)) = -\frac{1}{2}(y - m(X))^T (K + \sigma_y^2 I)^{-1}(y - m(X)) - \frac{1}{2}\log\Big(\det\Big(K + \sigma_y^2 I\Big)\Big)$$
$$-\frac{n_{\text{sim}}}{2}\log(2\pi).\tag{8}$$

### 2.3. Surrogate Model Quality

Before using a surrogate model, its quality must be quantified first. Two common methods are the coefficient of determination (Section 2.3.1) and cross validation (Section 2.3.2).

### 2.3.1. Coefficient of Determination

The coefficient of determination is a commonly used measure of the quality of a surrogate model. It is calculated by comparing the vector of deterministic model responses

$y$ with the vector of responses $\widetilde{y}$ predicted by the surrogate model. Various definitions can be found in the literature (cf. KvÅlseth [17]) as follows:

$$R_1^2 = 1 - \frac{\sum_i (y_i - \widetilde{y}_i)^2}{\sum_i (y_i - \overline{y})^2}, \tag{9}$$

$$R_2^2 = \frac{\sum_i (\widetilde{y}_i - \overline{y})^2}{\sum_i (y_i - \overline{y})^2}, \tag{10}$$

$$R_3^2 = \frac{\sum_i (\widetilde{y}_i - \overline{\overline{y}})^2}{\sum_i (y_i - \overline{y})^2}, \tag{11}$$

$$R_4^2 = 1 - \frac{\sum_i (\varepsilon_i - \overline{\varepsilon})^2}{\sum_i (y_i - \overline{y})^2}, \tag{12}$$

$$R_6^2 = \left( \frac{\mathrm{Cov}(y, \widetilde{y})}{\sqrt{\mathrm{Var}(y)} \cdot \sqrt{\mathrm{Var}(\widetilde{y})}} \right)^2 = \frac{\left( \sum_i (y_i - \overline{y_i}) \cdot \left( \widetilde{y}_i - \overline{\overline{y}} \right) \right)^2}{\sum_i (y_i - \overline{y_i})^2 \cdot \sum_i \left( \widetilde{y}_i - \overline{\overline{y}} \right)^2}, \tag{13}$$

$$R_7^2 = 1 - \frac{\sum_i (y_i - \widetilde{y}_i)^2}{\sum_i y_i^2}, \tag{14}$$

$$R_8^2 = \frac{\sum_i \widetilde{y}_i^2}{\sum_i y_i^2}. \tag{15}$$

Here, $\varepsilon_i = y_i - \widetilde{y}_i$ is the model error for the realization $i$. Furthermore, $\overline{y}$, $\overline{\overline{y}}$, and $\overline{\varepsilon}$ are the sample means of $y$, $\widetilde{y}$, and $\varepsilon$, respectively.

These sensitivity measures have different properties (cf. Prots [18]). To quantify the surrogate model quality, $R_1^2$ should be preferred. However, the $R_1^2$ value can be significantly reduced by a single or few data points that differs significantly from other observations (outliers). Therefore, an actual vs. predicted plot should always be analyzed to detect such points.

### 2.3.2. Cross Validation

Because the same data set is used for training the surrogate model and the computation of the coefficient of determination, it cannot be used to make statements about the predictive ability of a surrogate model. This is because, in engineering applications, the sample size is limited due to time and resource constraints, and splitting the data set into a training and test data set is also not feasible. For the best surrogate model quality, the whole data set is used. To make a statement about the predictive ability in this case, cross validation can be used.

In $k$-fold cross-validation, which is used in this paper, the data set is divided into $k$ subsets of approximately equal size. Then, the $k - 1$ of those subsets are used to train the surrogate model. The remaining subset is used to test the surrogate model. This process is repeated until each of those subsets was used for testing, thus resulting in a cross-validated coefficient of importance $R_{\mathrm{CV}}^2$.

Because the quality of a surrogate model is evaluated with independent data sets, $R_{\mathrm{CV}}^2$ provides a better measure of the predictive quality of a surrogate model than the coefficient of determination. Cross-validation is used only to assess the quality of the surrogate model. The final surrogate model is built using all available data points.

## 3. Sampling Methods

The creation of the sample is an essential step of the MCS and has direct influence on the surrogate model quality. In machine learning applications, the choice of the sampling methods is equally important, as a representative database is desired. This section compares different sampling methods, where the focus is on the impact on the surrogate model quality. For this, Section 3.1 demonstrates the impact of the selected sampling method on the surrogate model quality. Section 3.2 briefly discusses the existing sampling methods. In

Section 3.3, these methods are applied to two mathematical test functions, and the quality of the surrogate models are compared. Finally, Section 3.4 discusses how an existing sample created with oLHS can be extended.

### 3.1. Space-Filling Properties and Surrogate Model Quality

The creation of the sample is an essential step of the MCS and has direct influence on the surrogate model quality. This is illustrated in Figure 2 for a 1D test case. The dotted line represents the true model, and the black points are the sample points. As can be seen, the model is noisy and the realizations do not lie exactly on the dotted line. Using this sample, the surrogate model is created, as shown in red. As can be seen, using a poor sampling strategy can lead to a bad surrogate model (Figure 2a). The surrogate model quality can be increased by adding more points, especially in areas without any realizations. Another strategy would be to use a more uniform sampling in the beginning, as shown in Figure 2b. Here, the surrogate model almost perfectly matches the true model.

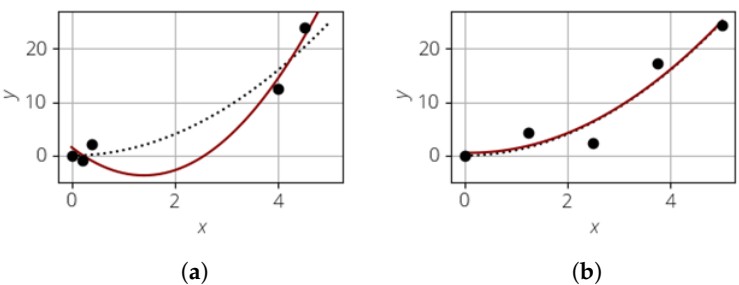

(a)                                                  (b)

**Figure 2.** Comparison of 1D surrogate models with non-uniform and uniform sampling, where $n_{sim} = 5$. (**a**) Non-uniform sampling. (**b**) Uniform sampling.

The problem statement can now be extended to a multi-dimensional case. Besides the marginal distributions, the correlation between the input variables must also be considered. It is usually described using the Spearman rank correlation coefficient $\tilde{r}$. However, there is no unique assignment between $\tilde{r}$ and the correlation structure. For example, both the point clouds shown in Figure 3 have a rank correlation coefficient close to 0 but are inherently different space-filling properties.

The point cloud in Figure 3a has poor space-filling properties. The large red ellipse marks an area in which no realizations are present and thus no information about the deterministic model behavior is available. On the other hand, the two realizations in the blue ellipse lie next to each other so that one realization does not provide any new information about the deterministic model compared to the other one. It would, therefore, be beneficial to move one realization into the area of the red ellipse. The sample shown in Figure 3b has good space-filling properties as no clusters or voids exist; therefore, it is desirable.

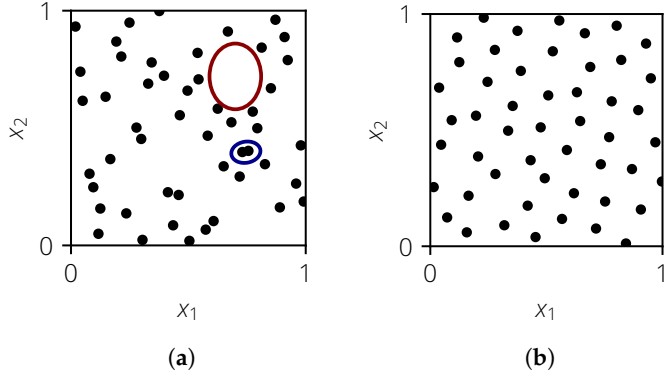

(a)                                                  (b)

**Figure 3.** Example of a 2D sample with poor and good space filling, where $n_{sim} = 50$. (**a**) Poor space filling. (**b**) Good space filling.

*3.2. Existing Sampling Methods*

In this section, different existing sampling methods are presented, namely simple random sampling (SRS, Section 3.2.1), Latin hypercube sampling (LHS, Section 3.2.2), optimized LHS (oLHS, Section 3.2.3), Latinized particle sampling (LPS, Section 3.2.4), and the Sobol sequence (Section 3.2.5). Furthermore, the methods for correlation control are discussed in Section 3.2.6.

### 3.2.1. Simple Random Sampling

In SRS, the random numbers for each input variable are generated independently without any constraints. First, $n_{\text{sim}}$ random numbers, which follow the uniform distribution $U(0;1)$ are generated as follows:

$$\boldsymbol{q} = \begin{pmatrix} q_1 & q_2 & \cdots & q_{n_{\text{sim}}} \end{pmatrix},$$

with $0 < q_i < 1$ for $i = 1, \ldots, n_{\text{sim}}$. To obtain the sample with the desired marginal distribution, the inverse cumulative distribution function (CDF) $F^{-1}(x)$ is applied to $\boldsymbol{q}$:

$$
\begin{aligned}
\boldsymbol{x} &= F^{-1}(\boldsymbol{q}) \\
&= \begin{pmatrix} F^{-1}(q_1) & F^{-1}(q_2) & \cdots & F^{-1}(q_{n_{\text{sim}}}) \end{pmatrix} \\
&= \begin{pmatrix} x_1 & x_2 & \cdots & x_{n_{\text{sim}}} \end{pmatrix}.
\end{aligned}
$$

The realizations of the individual input variables are then combined either randomly or a method for correlation control is applied.

### 3.2.2. Latin Hypercube Sampling

Latin hypercube sampling was introduced by McKay et al. [19], and it is a stratified approach. In comparison to SRS, the estimates of statistical quantities like the mean or standard deviation show a lower variation so that the same statistical significance can be achieved with a smaller sample size. A random point is placed in each of the intervals

$$[0;1/n_{\text{sim}}], \quad [1/n_{\text{sim}};2/n_{\text{sim}}], \quad \ldots \quad , \quad [(n_{\text{sim}}-1)/n_{\text{sim}};1],$$

thus resulting in the vector

$$\boldsymbol{q} = \begin{pmatrix} q_1 & q_2 & \cdots & q_{n_{\text{sim}}} \end{pmatrix},$$

with $(i-1)/n_{\text{sim}} < q_i < i/n_{\text{sim}}$ for $i = 1, \ldots, n_{\text{sim}}$. Again, the inverse CDF is applied to obtain the desired marginal distribution as follows:

$$
\begin{aligned}
\boldsymbol{x} &= F^{-1}(\boldsymbol{q}) \\
&= \begin{pmatrix} x_1 & x_2 & \cdots & x_{n_{\text{sim}}} \end{pmatrix}.
\end{aligned}
$$

Furthermore, special treatment can be conducted for $x_1$ and $x_{n_{\text{sim}}}$ to further reduce the variance of the statistical estimates (Huntington and Lyrintzis [20]).

In McKay et al. [19], the realizations of each input variable were combined randomly. As this can introduce spurious correlations, a correlation control algorithm like restricted pairing (RP) is usually applied.

### 3.2.3. Optimized Latin Hypercube Sampling

To remove the clusters and voids introduced by LHS, different approaches exist. Two commonly used methods are simulated annealing (cf. Morris and Mitchell [21], and Marrel [22]) and the enhanced stochastic evolutionary algorithm (Jin et al. [23]). Both algorithms swap the value of an input variable of two randomly selected realizations until a uniform space filling is reached (cf. Damblin et al. [24]). Note, the abbreviation oLHS refers to an LHS sample that has been optimized with respect to the space-filling properties and not the optimization method that was used to obtain it.

### 3.2.4. Latinized Particle Sampling

Latinized particle sampling was introduced by Prots et al. [25], and it considers the realizations as charged particles. Due to the repelling force, the realizations will be separated. This process is simulated in an iterative way. For this, three different force types are considered (Figure 4). The inner forces represent the repelling forces between any pair of two realizations (Figure 4a,b). The outer forces act from the walls onto the realizations (Figure 4c) so that they will remain within the sampling space. The frictional forces are used to dissipate energy to stop the movement of the realizations (Figure 4d). In an iterative way, the positions and velocities are updated until a force equilibrium is reached. Afterward, the sample is Latinized to obtain the desired marginal distributions. Finally, a correlation control algorithm can be applied to obtain the desired target correlation. More details can be found in Prots [18].

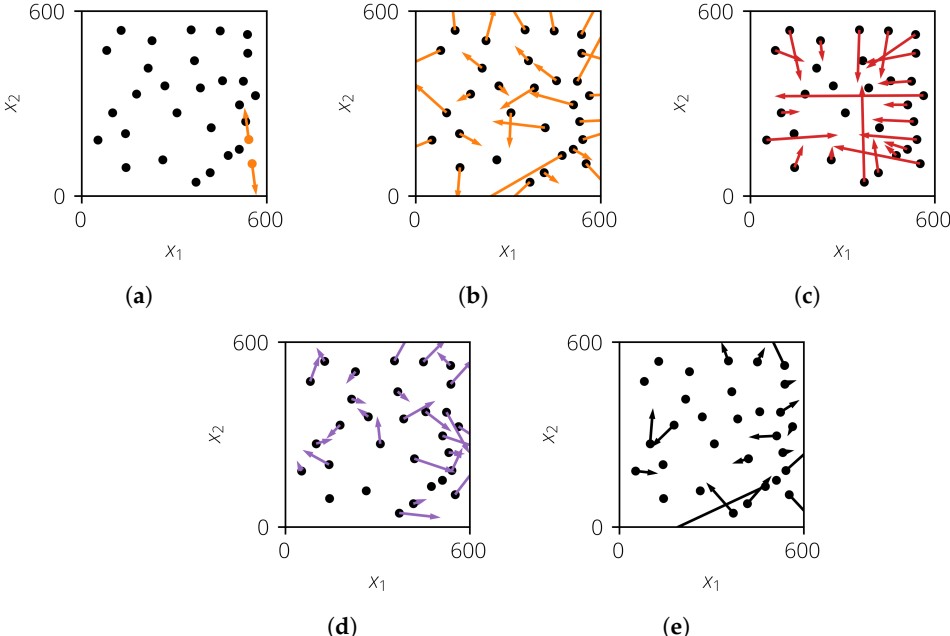

**Figure 4.** Calculation of forces for one iteration step. (**a**) The inner force (orange) between two realizations; (**b**) the resulting inner force (orange) of all realizations; (**c**) the resulting outer force (red) of all realizations; (**d**) the resulting frictional force (violet) of all realizations; and (**e**) the resulting total force (black) of all realizations.

### 3.2.5. Sobol Sequence

The Sobol sequence (also called the $LP_\tau$ sequence Sobol [26]) is a sampling method where the realizations are generated in a deterministic way. It is based on the one-dimensional van der Corput sequence [27] $\Phi_2(k)$. Using different so-called direction numbers, a multi-dimensional sample can be created. More details are given by Lemieux [28].

### 3.2.6. Correlation Control

The discussed sampling methods do not have a means of setting a desired target correlation matrix. However, in engineering applications, this is important as the input variables are usually correlated. For this, correlation control algorithms like restricted pairing (RP) can be used (Iman and Conover [29]). Here, a data matrix $X$ is transformed into the matrix $X^*$ with the desired target Spearman rank correlation matrix $\widetilde{T}$. The following steps are performed (cf. Dandekar et al. [30]):

1. Compute the lower triangular matrix $P$ so that $PP^T = \widetilde{T}$ holds.
2. Compute the rank correlation matrix $\widetilde{C}$ of the matrix $X$.
3. Compute the lower triangular matrix $Q$ so that $QQ^T = \widetilde{C}$ holds (e.g., using the Cholesky decomposition, Benoit [31]).

4. Compute the matrix $S = PQ^{-1}$.
5. Compute the matrix $R = XS^T$.
6. Replace the values of $R$ by their ranks in the corresponding column.
7. Sort the values of $X$ according to their ranks in $R$ to form the matrix $X^*$.

The result of the RP depends significantly on the initial matrix $X$. For LHS, it can be scrambled multiple times before applying RP. For oLHS, LPS, and the Sobol sequence, the initial matrix $X$ remains unchanged to prevent the space-filling properties.

### 3.3. Comparison of Sampling Methods

The sampling methods are now compared with respect to the resulting quality of the surrogate models. For this, the following steps are performed:

1. Create sample $X_{\text{train}}$.
2. Evaluate $y_{\text{train}} = f(X_{\text{train}})$.
3. Create surrogate model $\widetilde{f}(x)$ using $X_{\text{train}}$ and $y_{\text{train}}$.
4. Evaluate surrogate model quality.

The surrogate model quality is quantified using a separate test data set $y_{\text{test}} = f(X_{\text{test}})$ as follows:

$$R^2_{1,\text{test}} = 1 - \frac{\sum_{i=1}^{n}(y_{\text{test},i} - \widetilde{y}_{\text{test},i})^2}{\sum_{i=1}^{n}(y_{\text{test},i} - \overline{y}_{\text{test}})^2}, \tag{16}$$

with $\widetilde{y}_{\text{test},i} = \widetilde{f}(x_{\text{test},i})$. This procedure is repeated 250 times to obtain a distribution of $R^2_{1,\text{test}}$. The test data are the same for all repetitions and are created with LHS for simplicity ($n_{\text{test}} = 2000$).

Because of the repetitions, it is not possible to perform this analysis for engineering test cases. Therefore, two mathematical test functions will be analyzed.

#### 3.3.1. Sasena Test Function

For this test, a 2D mathematical function, introduced by Ben Salem and Tomaso [32], is analyzed. It is defined as

$$y(x) = 2 + 0.01(x_2 + x_1^2)^2 + (1 - x_1)^2 + 2(2 - x_2)^2 + 7\sin(0.5x_1)\sin(0.7x_1x_2) + \varepsilon, \tag{17}$$

with $x_1, x_2 \sim U(0; 5)$. With $\sigma_\varepsilon \sim N(0; \sigma_\varepsilon)$, a normally distributed noise term is considered. The analysis is performed for $n_{\text{sim}} = 50$. The surrogate model is a GP.

Two test cases are analyzed. In the first test case, no noise is considered and the parameter $\sigma_y$ is set to $10^{-5}$. Figure 5a shows the distribution of $R^2_{1,\text{test}}$. If the distribution lies more to the right, then this means that the corresponding sampling method yields better surrogate models in a statistical sense. In this particular case, the curves for oLHS and LPS match and are more to the right compared to the curves of LHS and LSOB-S, such that the former two sampling methods are superior.

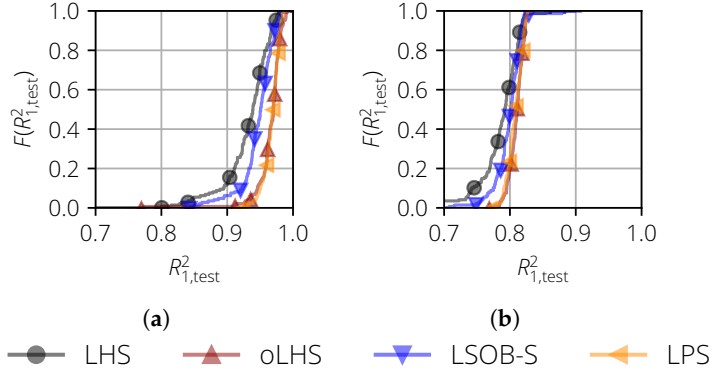

**Figure 5.** Empirical cumulative distribution function of $R^2_{1,\text{test}}$ for the surrogate models of the Sasena test function. (**a**) Noise-free case and fixed $\sigma_y = 10^{-5}$; (**b**) noisy case ($\sigma_\varepsilon = 2.5$) and free $\sigma_y$.

The same behavior can be observed for the second case, where a noise with $\sigma_\varepsilon = 2.5$ was considered. During the surrogate model training, the parameter $\sigma_y$ was set free and was therefore also trained. Due to the noise, the quality of the surrogate model decreased for all sampling methods Nevertheless, oLHS and LPS yielded the best surrogate models.

### 3.3.2. Oakley & O'Hagan Test Function

In the second test case, a 15D mathematical test function, introduced by Oakley and O'Hagan [33], was analyzed. It is defined as

$$y(\boldsymbol{x}) = \boldsymbol{a}_1^T \boldsymbol{x} + \boldsymbol{a}_2^T \sin(\boldsymbol{x}) + \boldsymbol{a}_3^T \cos(\boldsymbol{x}) + \boldsymbol{x}^T \boldsymbol{M} \boldsymbol{x}, \tag{18}$$

with $\boldsymbol{x} \in \mathbb{R}^{n_{\text{sim}} \times 15}$. The values for $\boldsymbol{a}_1$, $\boldsymbol{a}_2$, $\boldsymbol{a}_3$, and $\boldsymbol{M}$ are given by Oakley [34]. The sample size is $n_{\text{sim}} = 80$.

For this test case, the results of the different types of surrogate models were analyzed. For a polynomial surrogate model (Figure 6a), the largest differences could be seen. The best surrogate models were obtained by oLHS, followed by LPS, LSOB-S, and LHS.

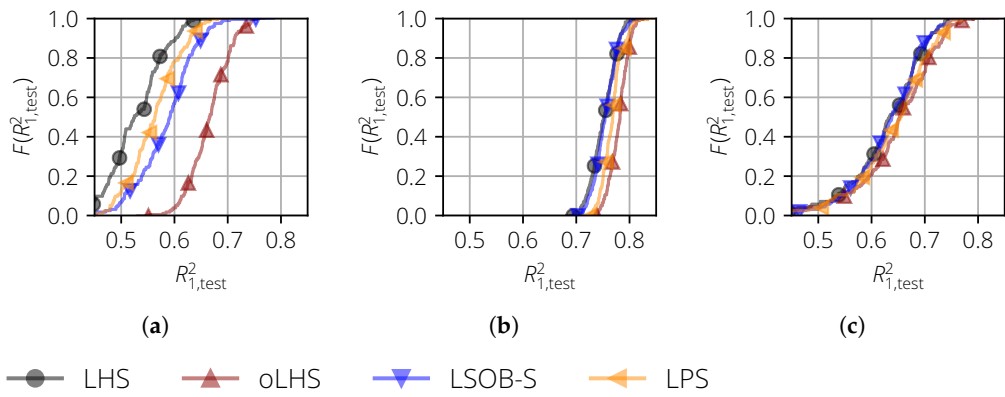

**Figure 6.** Empirical cumulative distribution function of $R^2_{1,\text{test}}$ for the surrogate models of the Oakley and O'Hagan test function and the uncorrelated input variables. (**a**) Polynomial; (**b**) the Gaussian process and one-dimensional length scale; and (**c**) the Gaussian process and multi-dimensional length scale.

For a GP with a 1D-length scale (Figure 6b), the order is the same. However, the differences between the sampling methods become marginal. For a GP with a multi-dimensional-length scale (Figure 6c), there was virtually no difference between the sampling methods.

### 3.3.3. Summary

As demonstrated in this section, the performance of sampling methods with respect to the quality of surrogate models depends on the selected test case and surrogate model. However, when there was a difference between the sampling methods, then oLHS performed the best, especially for the high-dimensional test case. Therefore, oLHS is preferred when a high surrogate model quality is required.

### 3.4. Extension of oLHS

Another advantage of oLHS is that fixed realizations can be used. On example are predefined points that should be included in the sampling. This is shown in Figure 7a, where the red points were predefined. They can represent the nominal design or other combinations of input variables of interest. The remaining points were added using the oLHS approach, and the entire sample exhibited good space-filling properties.

Furthermore, this approach can be used to extend an existing sample to increase the sample size. This is shown in Figure 7b. The red points represent the initial sample, which was created with oLHS and has good space-filling properties ($n_{\text{sim}} = 40$). Then, another 40 realizations were added (black points). As can be seen, the space-filling properties are

still good. In comparison, Figure 7c shows the extension of the same sample using extended LHS (eLHS, Schmidt et al. [35]), where the space-filling properties are not tracked and the resulting sample has inferior space-filling properties in comparison to the extension with oLHS.

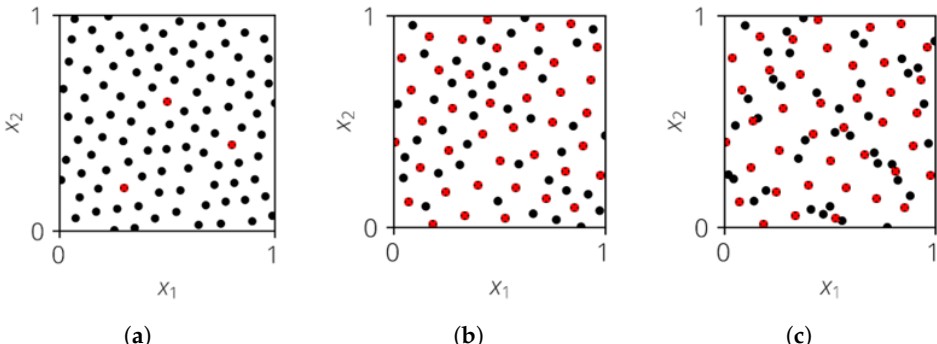

(**a**)                    (**b**)                    (**c**)

**Figure 7.** Example of oLHS (black and red) with fixed realizations (red). (**a**) Predefined points; (**b**) an extension of a sample; (**c**) and a comparison with eLHS.

## 4. Sensitivity Analysis

### 4.1. Basics of Sensitivity Analysis

Sensitivity analysis is "*the study of how uncertainty in the output of a model (numerical or otherwise) can be apportioned to different sources of uncertainty in the model input*" (Saltelli et al. [36] (p. 45)). An overview of sensitivity analysis methods can be found, for example, in Saltelli et al. [37].

In engineering applications, a global sensitivity analysis is performed, where the influence of the input variables in the entire definition range is quantified. Examples of global sensitivity measures are the Sobol indices (Sobol and Levitan [38]), as well as the Shapley values (Shapley [39]), which are discussed in the following.

The definition of both sensitivity measures is quite complex and not intuitive without experience. Therefore, an illustrative example is introduced, which is then used to easily describe the different sensitivity measures. For this, the restaurant order in Table 1 is analyzed. There, Alice, Bob, and Charlie each order something to eat and drink. Furthermore, they give a tip at the end. Some orders can be assigned to one single person, and some orders are shared between two people.

The behavior of the different sensitivity measures on a mathematical test function is shown in Prots [18] and Prots et al. [40]. There, the effect of a correlation is demonstrated, which cannot be considered in the simple example of Table 1.

**Table 1.** Illustrative example for sensitivity analysis.

|         | Meal          | Drink       | Tip       |
| ------- | ------------- | ----------- | --------- |
| **Alice**   |               | Coke, $3    |           |
| **Bob**     | Paella, $24   |             | Tip, $6   |
| **Charlie** | Burger, $15   | Wine, $16   |           |

#### 4.1.1. Sobol Indices

With the Sobol indices, the influence of a subset of one or more input variables on the variance of $f(\boldsymbol{x})$ can be quantified. In the scope of this paper, the following types are discussed:

- First-order Sobol sensitivity indices;
- Higher-order Sobol sensitivity indices;
- Total-effect Sobol sensitivity indices.

These were originally defined for uncorrelated inputs. However, Kucherenko et al. [41] has shown that these definitions are also valid for correlated input variables.

First-Order Sobol Sensitivity Indices

The first-order Sobol sensitivity index $S_i$ (also called the main sensitivity index MSI) is defined as the partial variance contributed by the input variable $x_i$, which is normalized to the total variance as follows:

$$S_i = \frac{\text{Var}(\mathbb{E}(f(\boldsymbol{x})|x_i))}{\text{Var}(f(\boldsymbol{x}))} \,. \tag{19}$$

The alternative definition

$$S_i = 1 - \frac{\mathbb{E}(\text{Var}(f(\boldsymbol{x})|x_i))}{\text{Var}(f(\boldsymbol{x}))} \tag{20}$$

can be derived from the law of total variance

$$\text{Var}(f(\boldsymbol{x})) = \text{Var}(\mathbb{E}(f(\boldsymbol{x})|x_i)) + \mathbb{E}(\text{Var}(f(\boldsymbol{x})|x_i)) \,. \tag{21}$$

The number of MSIs is equal to the number of input variables $n_d$.

Applying the MSIs to the example of Table 1 provides the values

$$S(A) = USD\ 3 \,, \qquad S(B) = USD\ 0 \,, \qquad S(C) = USD\ 15 \,,$$

where $S(A)$, $S(B)$, and $S(C)$ are the MSIs of Alice, Bob, and Charlie, respectively. As can be seen, they equal the orders that were made by this person only. This means, that the MSI of Bob is equal to 0 because he only has shared orders. Therefore, quantifying the importance of Bob solely based on the MSI is misleading as it does not consider interaction effects.

This problem can be generalized: the MSI of an input variable can be 0, even if it has a functional influence on the function. Therefore, it is not sufficient to only consider the MSIs.

Higher-Order Sobol Sensitivity Indices

The higher-order Sobol sensitivity index $S_{i_1,\dots,i_s}$ (also called the interaction sensitivity index, ISI) is defined as the partial variance caused by two or more input variables $i_1,\dots,i_s$, and it is normalized to the total variance. The second-order Sobol sensitivity index is calculated as (Saltelli et al. [37])

$$S_{i_1,i_2} = \frac{\text{Var}(\mathbb{E}(f(\boldsymbol{x})|x_{i_1}, x_{i_2}))}{\text{Var}(f(\boldsymbol{x}))} - S_{i_1} - S_{i_2} \,. \tag{22}$$

Third-order Sobol sensitivity indices can also be analogously calculated in this manner.

The number of higher-order Sobol sensitivity indices is $2^{n_d} - 1 - n_d$, which rise exponentially with increasing $n_d$. Therefore, only the first-order Sobol sensitivity indices are usually calculated. For small $n_d$, the second-order Sobol sensitivity indices can also be computed, where the number of indices is $n_d \cdot (n_d - 1)/2$. However, if there is an interaction effect that is caused by three or more input variables, this effect is then overlooked when only calculating first-order and second-order Sobol sensitivity indices.

For the example of Table 1, the ISIs were

$$S(A,B) = USD\ 24 \,, \qquad S(A,C) = USD\ 0 \,, \qquad S(B,C) = USD\ 16 \,, \qquad S(A,B,C) = USD\ 6 \,.$$

The ISI of Alice and Bob is USD 24, as their meal order is the only one that both of them share exclusively. Because there is no order where Alice and Charlie solely contribute, their ISI is USD 0. The tip is included in the third-order ISI.

As already mentioned, the number of ISIs is very large in engineering applications, so they they are not usually computed.

Total-Effect Sobol Sensitivity Indices

The total-effect Sobol sensitivity index (TSI) $S_{T,i}$ of the input variable $x_i$ is the sum of the MSI $S_i$ and all ISIs that include interaction effects with $x_i$ as follows:

$$S_{T,i} = \sum_{\mathcal{S} \subseteq \mathcal{M} : i \in \mathcal{S}} S_{\mathcal{S}} \,. \tag{23}$$

It can alternatively be calculated by (Saltelli et al. [37])

$$S_{T,i} = \frac{\mathbb{E}(\text{Var}(f(\boldsymbol{x})|\boldsymbol{x}_{\sim i}))}{\text{Var}(f(\boldsymbol{x}))} \tag{24}$$

$$= 1 - \frac{\text{Var}(\mathbb{E}(f(\boldsymbol{x})|\boldsymbol{x}_{\sim i}))}{\text{Var}(f(\boldsymbol{x}))} \,, \tag{25}$$

where $f(\boldsymbol{x})$ is conditioned by all input variables except $x_i$ (written as $\boldsymbol{x}_{\sim i}$). The number of TSIs is equal to the number of input variables $n_d$.

Applying the TSI to the example of Table 1, the TSIs are

$$S_T(A) = USD\ 24 + USD\ 3 + USD\ 6 = USD\ 33 \,,$$
$$S_T(B) = USD\ 24 + USD\ 16 + USD\ 6 = USD\ 46 \,,$$
$$S_T(B) = USD\ 15 + USD\ 16 + USD\ 6 = USD\ 37 \,.$$

The TSI considers the total contribution of a given input variable. Thus, if $S_T$ is equal to 0, it might be concluded that this input variable has no contribution to the variance of $f(\boldsymbol{x})$. This is true for uncorrelated input variables (Saltelli et al. [37]). However, for correlated input variables, the TSI can become 0 for an input variable even if it has a functional influence.

### 4.1.2. Shapley Values

Shapley values (Shapley [39]) are a concept in cooperative game theory that fairly distribute the payoff generated by a coalition among its players by taking into account the contributions of each member and the interactions among them. Their definition is similar to TSIs. However, the ISIs are distributed equally to the corresponding input variables as follows:

$$Sh_i = \sum_{\mathcal{S} \subseteq \mathcal{M} : i \in \mathcal{S}} \frac{S_{\mathcal{S}}}{|\mathcal{S}|} \,, \tag{26}$$

where $|\mathcal{S}|$ is the cardinality of $\mathcal{S}$. The number of Shapley values is equal to the number of input variables $n_d$.

Applying the Shapley values to the example of Table 1 yields

$$Sh(A) = \frac{USD\ 24}{2} + USD\ 3 + \frac{USD\ 6}{3} = USD\ 17 \,,$$
$$Sh(B) = \frac{USD\ 24}{2} + \frac{USD\ 16}{2} + \frac{USD\ 6}{3} = USD\ 22 \,,$$
$$Sh(C) = USD\ 15 + \frac{USD\ 16}{2} + \frac{USD\ 6}{3} = USD\ 25 \,.$$

As can be seen, the shared orders are distributed equally to the corresponding persons. In this particular example, the Shapley value is equal to the amount each person would have to pay.

The Shapley values are a suitable way through which to analyze a problem with interactions. In general, this applies to both functional and correlation interactions. However, if $x_1$ is an input variable, which does not have a function impact on $f(\boldsymbol{x})$ but has a high correlation to an input variable $x_2$ with a high impact, it will still obtain a high Shapley value. Furthermore, the Shapley value of $x_2$ will decrease. In theory, one could add many

such correlated variables without a functional influence so that the Shapley value of $x_2$ will be reduced even further. It is, therefore, critical to identify input variables without functional influence.

### 4.2. Modified Coefficient of Importance

As discussed, the Sobol indices and Shapley values are not sufficient to analyze a system with correlated input variables, as both can become 0 for input variables with functional influence. Hence, it is not reliably possible to identify input variables without functional influence. Therefore, an additional sensitivity measure is used, which is the modified coefficient of importance (mCoI, Prots [18], and Prots et al. [40]).

The mCoI is based on the coefficient of importance introduced by Bucher [42], and it combines this concept with the idea of the quantification of variable importance by a random forest (Breiman [43]). First, a test sample $X_{\text{test}}$ is created and evaluated, thereby yielding $y_{\text{test}} = f(X_{\text{test}})$. Then, for each input variable $x_i$, a copy of $X_{\text{test}}$ is created. Unlike for the CoI, the values of $x_i$ are not set constant but are permuted, thus yielding $X_{\text{test}}^i$. Evaluating this data set gives $y_{\text{test}}^i = f(X_{\text{test}}^i)$. Now, the *mCoI* for the input variable $x_i$ is calculated as

$$mCoI_{x_i} = 1 - R_4^2(y_{\text{test}}, y_{\text{test}}^i),$$
(27)

with $R_4^2$ as the coefficient of determination from Equation (12).

If $x_i$ is an input variable with a functional influence, then this means that, after permuting, $y_{\text{test}}^i$ is much different than $y_{\text{test}}$ so that the $R_4^2$ is low and the mCoI is close to 1.0. On the other hand, if $x_i$ has no functional influence, then $y_{\text{test}}$ and $y_{\text{test}}^i$ are very similar and $R_4^2$ is close to 1.0, thus resulting in a low mCoI value.

For a better interpretation of the mCoI, the values are normalized so that

$$\sum_i^{n_d} mCoI_{x_i} = 1.$$
(28)

This process is now repeated $n_{\text{repeats}}$ times (e.g., $n_{\text{repeats}} = 500$), and the median value is used as the mCoI.

Because many function evaluations are required for the mCoI, $f(X)$ is usually represented by a surrogate model. For a result with statistical significance, a high-surrogate-model quality is required. For the sensitivity analysis, all three measures (Sobol indices, Shapley values, and mCoI) are computed. The mCoI is used to identify the input variables without functional influence. The remaining sensitivity measures can then be used.

## 5. Application to Turbomachinery

In this final section, the sensitivity analysis is performed for a turbomachinery test 444 case. First, the test case is presented in Section 5.1. Then, the probabilistic setup is shown in Section 5.2 and the results of the sensitivity analysis are discussed in Section 5.3.

### 5.1. Turbomachinery Test Case

In the turbomachinery test case, the post-service compressor blades of an industrial mid-stage rotor of a high-pressure compressor, which are subject to manufacturing variability and wear, were analyzed. The optical measurements were performed with an ATOS Scanbox 5108 from GOM. From the internal investigations conducted at the Chair of Turbomachinery and Flight Propulsion of the Technische Universität Dresden, the measurement accuracy was estimated to be about $20\,\mu$m. A total of 77 optically measured blades were obtained from this analysis.

The variability of the blades was parameterized using the approach introduced by Lange et al. [44] and Heinze [45]. There, the profile parameters similar to the NACA parameters were used to describe the camber and thickness distribution to represent the manufacturing variability. Additionally, positional parameters were used to describe a

change in the position of the blade. These parameters are visualized in Figure 8 and listed in Table 2.

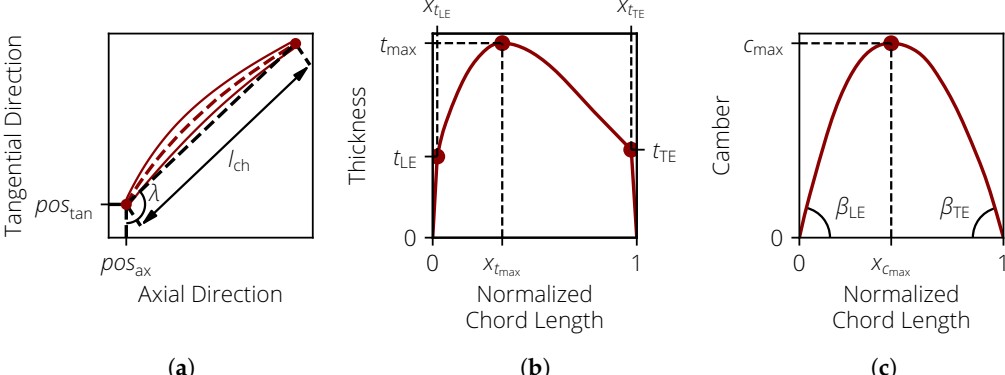

**Figure 8.** Visualization of the blade profile parameters. (**a**) Profile Parameters. (**b**) Thickness parameters. (**c**) Camber parameters.

**Table 2.** The geometric blade profile parameters (cf. Lange et al. [1] (Table 1)).

| Symbol | Variable |
| --- | --- |
| $pos_{ax}$, $pos_{tan}$ | Axial and tangential positions of the section outline at the leading edge |
| $\lambda$ | Stagger angle |
| $l_{ch}$ | Chord length |
| $t_{LE}$, $t_{TE}$ | Thickness of the leading edge and trailing edge |
| $x_{t_{LE}}$, $x_{t_{TE}}$ | Position of $t_{LE}$ and $t_{TE}$ on the chord |
| $t_{max}$ | Maximum thickness of the profile |
| $x_{t_{max}}$ | Position of $t_{max}$ on the chord |
| $\beta_{LE}$, $\beta_{TE}$ | Angle of the camber line at the leading edge and trailing edge |
| $c_{max}$ | Maximum camber of the profile |
| $x_{c_{max}}$ | Position of $c_{max}$ on the chord |

To obtain the distribution of the profile parameters, a population of blades was analyzed. In the first step, the profile parameters are obtained for each blade. For this, the profile contours (sections) are extracted on different span positions as the intersection between the blade body and the rotating body that results from a streamline. At each span position, the 14 profile parameters are retrieved, thereby resulting in the parameter matrix

$$\boldsymbol{P}_i = \left[ \boldsymbol{p}_{i,1}, \boldsymbol{p}_{i,2}, \ldots, \boldsymbol{p}_{i,n_{\text{sec}}} \right]^T \in \mathbb{R}^{n_{\text{sec}} \times 14} , \tag{29}$$

where $i$ is the index of the analyzed blade, $\boldsymbol{p}_{i,j}$ is the parameter vector for section $j$, and $n_{\text{sec}}$ is the number of sections (here $n_{\text{sec}} = 48$).

In the next step, the delta parameters $\Delta \boldsymbol{P}_i = \boldsymbol{P}_{\text{ref}} - \boldsymbol{P}_i$ are obtained, where $\boldsymbol{P}_{\text{ref}}$ is the parameter matrix of a reference model. In the original approach by Lange et al. [44], the nominal design (ND) is used as the reference model. Heinze et al. [45] introduced a so-called median model $\overline{\boldsymbol{P}}_{\text{ref}}$, which was obtained by analyzing the entire population of the blades. For each profile parameter at each section, the median of the population was used to create $\overline{\boldsymbol{P}}_{\text{ref}}$. Using the median model, the radial distribution of the delta parameters is more similar, which simplifies their description.

The main driving forces of the geometric variability for the analyzed blade population are manufacturing variability and wear. Therefore, the parameters are strongly correlated in the radial direction, so that the $n_{\text{av}} \leq n_{\text{sec}}$ averaging domains can be used in

a radial direction for dimensionality reduction (cf. Lange et al. [46]), thus resulting in the parameter set

$$\Delta \overline{P}_i = \left[ \Delta \overline{p}_{i,1}, \Delta \overline{p}_{i,2}, \ldots, \Delta \overline{p}_{i,n_{av}} \right]^T \in \mathbb{R}^{n_{av} \times 14} \text{ with } n_{av} \leq n_{sec}. \tag{30}$$

Usually, $n_{av} = 1$ is used to keep the number of input variables small. To demonstrate the effect of correlations for the sensitivity analysis, $n_{av} = 4$ was also used. The correlation matrices are shown in Figure 9. For $n_{av} = 4$, each block represents an averaging section.

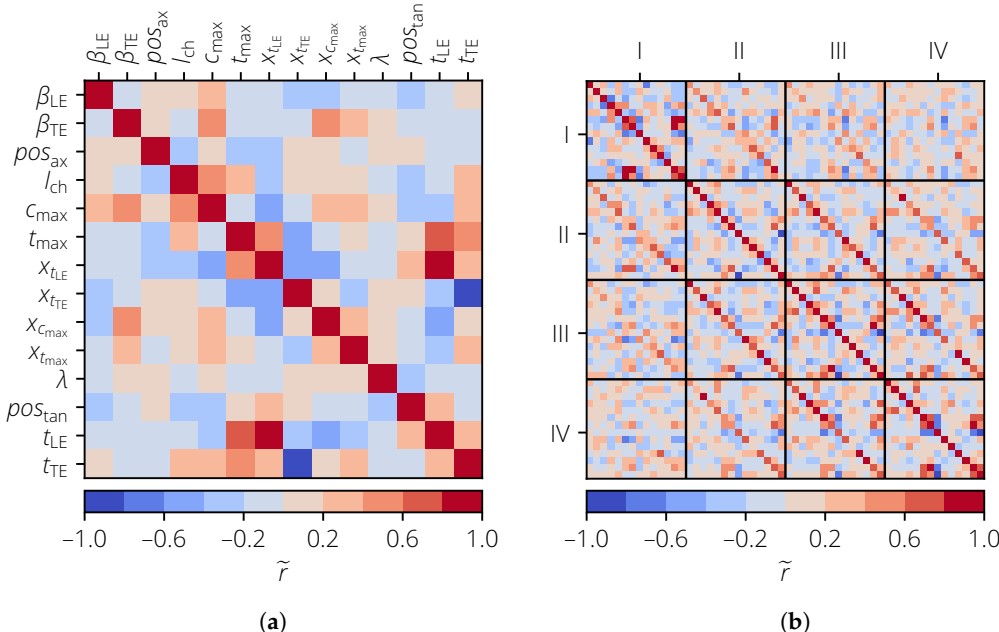

(a)  (b)

**Figure 9.** Correlation matrix of the profile parameters for the turbomachinery test case. (**a**) $n_{av} = 1$. (**b**) $n_{av} = 4$.

The result value of interest is the isentropic efficiency $\eta_{is}$, which is defined as

$$\eta_{is} = \frac{\Delta h_{is}}{\Delta h}, \tag{31}$$

with $\Delta h_{is}$ and $\Delta h$ as the changes in the specific enthalpy of the isentropic and real process, respectively. The 3D CFD calculations were performed with the Rolls-Royce proprietary HYDRA CFD system using a stator–rotor–stator configuration. The geometry of the vanes is kept constant to focus only on the effect of the blades. The flow domain is discretized using the tool PADRAM, thus resulting in an O-H-grid mesh of approximately $4.6 \times 10^6$ nodes. The flow field is calculated with stationary Reynolds-averaged Navier-Stokes equations with the Spalart-Allmaras turbulence model. Boundary conditions are specified as radial profiles of the total temperature, total pressure, radial and tangential flow angles, and the Spalart variable at the inlet. A flow capacity is specified at the outlet of the domain. The analyzed operating point is at the design point speed line near peak efficiency. The flow field around the blade is visualized in Figure 10. There, the wall shear stress (an indicator for the boundary layer state) and the Mach number are shown.

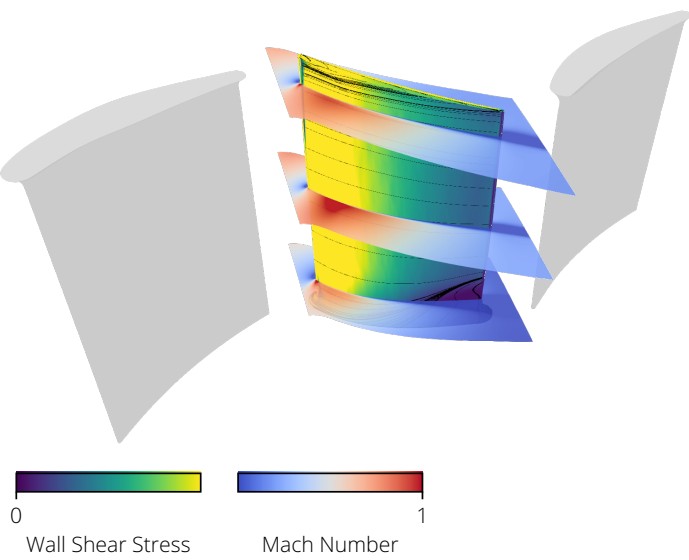

**Figure 10.** The flow field around the blade.

### 5.2. Probabilistic Setup

To perform sensitivity analysis, a surrogate model is needed. For each $n_{av}$, a separate MCS is performed since each case has a different set of input variables. The sample sizes are $n_{sim} = \{80, 320\}$ for $n_{av} = \{1, 4\}$, respectively. All samples are created using oLHS in combination with RP. For all input variables, a normal distribution is assumed.

A surrogate model is then created for each MCS. For all three values of $n_{av}$, a first-order polynomial surrogate model is created. The actual vs. predicted plots are shown in Figure 11. The surrogate models can predict $\eta_{is}$ well, and no clear outliers can be seen. This is also reflected in the high values of $R_1^2 > 0.98$ and $R_{1,CV}^2 > 0.98$. This means that these surrogate models can be used for further analysis.

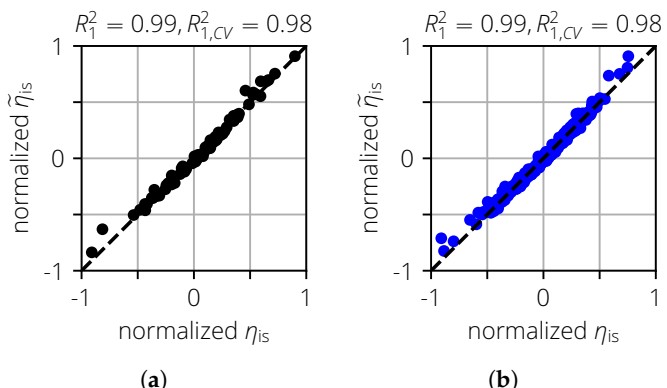

**Figure 11.** Actual vs. predicted plots for the turbomachinery test case. (**a**) $n_{av} = 1$. (**b**) $n_{av} = 4$.

### 5.3. Sensitivity Analysis

The results of the sensitivity analysis are shown in Figure 12. For a better comparability of the results, the parameters were grouped for $n_{av} = 4$. A group consists of the same profile parameters in the different averaging domains. Based on the mCoI, $t_{LE}$, $c_{max}$, and $x_{t_{LE}}$ can be identified as the most important input variables for the isentropic efficiency $\eta_{is}$. All other input variables have a significantly smaller mCoI value, which are close to 0.

For the Shapley values and first-order Sobol indices, there is a good agreement for most input variables. For $\beta_{LE}$, on the other hand, larger differences are present. For example, $Sh(\beta_{LE}) = 0.01$ and $S_1(\beta_{LE}) = 0.01$ for $n_{av} = 1$. For $n_{av} = 4$, however, both values increased ($Sh(\beta_{LE}) = 0.05$ and $S_1(\beta_{LE}) = 0.15$). This must be caused by the correlations of $\beta_{LE}$ with other input variables, since functional interactions are not modeled in the used

first-order polynomial. For $n_{av} = 4$, the first-order Sobol sensitivity index increased again ($S_1(\beta_{LE}) = 0.29$). A similar behavior could be observed for $\beta_{TE}$. In all three cases, however, the corresponding mCoI values for both $\beta_{LE}$ and $\beta_{TE}$ were close to 0; as such, both profile parameters were identified as one without functional influence in all three cases.

In this example, the advantage of the mCoI over the total-effect Sobol index was further illustrated. Due to the correlations between the profile parameters (especially for $n_{av} = 4$), the significance of $S_T$ decreases. Thus, input variables without functional influence cannot be reliably determined. For example, $S_T(x_{t_{LE}}) = 0.01$ for $n_{av} = \{2, 4\}$. The mCoI, on the other hand, clearly shows that $x_{t_{LE}}$ has a non-negligible influence on $\eta_{is}$.

The identification of important and unimportant input variables is an important step for a further analysis of the compressor blade. For example, if a robust optimization is performed, input variables without functional influence can be removed from the analysis to reduce the search space. But it can also have practical applications. For example, the manufacturing process can be changed in such a way that the variability for variables without functional influence is increased in order to save costs during production.

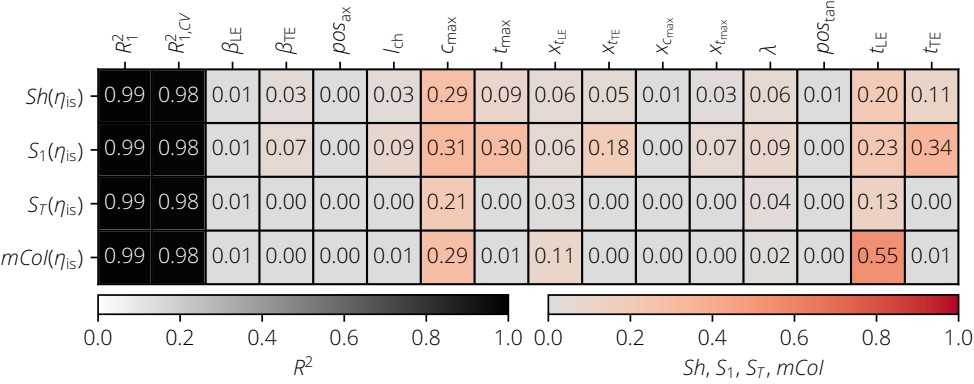

(**a**)

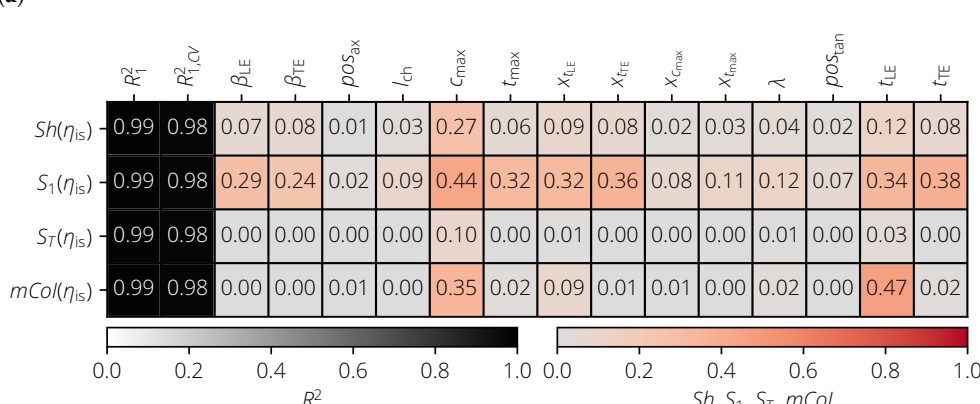

(**b**)

**Figure 12.** Results of the sensitivity analysis of the turbomachinery test case for different numbers of averaging domains. (**a**) $n_{av} = 1$. (**b**) $n_{av} = 4$.

## 6. Summary

In this paper, different strategies of probabilistic analyses were discussed. These allow for taking the uncertainty in geometry and boundary conditions into account. Monte Carlo simulation (MCS) is a popular method but requires an increased numerical effort. Hence, even small improvements in probabilistic methods can have a large impact.

The first part of this paper discussed and analyzed different sampling methods. There, the focus was laid on the quality of surrogate models because they can be evaluated quickly and can serve as a replacement for a complex numerical model. The discussed methods were Latin hypercube sampling (LHS), optimized LHS (oLHS), the Sobol sequence, and Latinized particle sampling (LPS). They exhibit different properties with respect to the

space filling of a sampling space. For different mathematical test cases, these methods were used to create surrogate models. The best ones were obtained with oLHS, which should therefore be used if a high surrogate model quality is desired.

In the second part of this paper, methods for sensitivity analysis were discussed. The Sobol indices and Shapley values were explained on an intuitive example. However, since, for correlated input variables, these are not sufficient to identify input variables without functional influence, the modified coefficient of importance (mCoI) was discussed as an additional sensitivity measure.

In the third part, these methods were applied to a turbomachinery test case. There, the blades of a compressor row were subject to geometric variability so that the isentropic efficiency was also subject to variability. To perform the sensitivity analysis, a MCS was first performed to create a surrogate model using oLHS as the sampling method. Then, a sensitivity analysis was performed for the different sets of input variables. There, the advantage of the mCoI was demonstrated as it could reliably identify the input variables without functional influence. It was shown that, for this particular test case, the variability of the maximum camber and thickness of the leading edge had the largest effect on the variability of the isentropic efficiency.

**Author Contributions:** Conceptualization, A.P. and M.V.; methodology, A.P.; software, A.P. and M.V.; validation, M.V.; formal analysis, A.P.; investigation, A.P.; resources, M.V. and R.M.; data curation, A.P.; writing—original draft preparation, A.P.; writing—review and editing, M.V. and R.M.; visualization, A.P.; supervision, M.V. and R.M.; project administration, M.V. and R.M.; funding acquisition, M.V. and R.M. All authors have read and agreed to the published version of the manuscript.

**Funding:** The research was funded by the German Federal Ministry of Economic Affairs and Climate Action under grant number 20X1909H.

**Data Availability Statement:** Data is not available due to confidentiality reason.

**Acknowledgments:** This work was conducted as part of the joint research program DIGIfly in the frame of the Luftfahrtforschungsprogramm (LuFo) which was funded by the German Federal Ministry of Economic Affairs and Climate Action. The authors gratefully acknowledge Rolls-Royce Deutschland Ltd & Co KG for their support and permission to publish this paper. Computations were performed on a high-performance computing system at the Center for Information Services and High Performance Computing (ZIH) at TU Dresden. The authors thank the ZIH for their generous allocation of computational resources.

**Conflicts of Interest:** The authors declare no conflicts of interest.

## Abbreviations

The following abbreviations are used in this manuscript:

| | |
|---|---|
| CoI | Coefficient of importance |
| eLHS | Extended Latin hypercube sampling |
| GP | Gaussian process |
| ISI | higher-order Sobol sensitivity index |
| LHS | Latin hypercube sampling |
| LPS | Latinized particle sampling |
| mCoI | Modified coefficient of importance |
| MCS | Monte Carlo simulation |
| MSI | First-order Sobol sensitivity index |
| ND | Nominal design |
| oLHS | Optimized Latin hypercube sampling |
| RP | Restricted pairing |
| SRS | Simple random sampling |
| TSI | Total-effect Sobol sensitivity index |

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
