# Peer review of "Improvements in Probabilistic Strategies and Their Application to Turbomachinery"

_aerospace, doi:10.3390/aerospace11050355_

Round 1

Reviewer 1 Report

Comments and Suggestions for Authors

Solid paper: good introduction and first and second part. I'm not fully convinced by the third part on the turbomachinery application, and I think it requires further clarification for a journal publication.

1) The comparison you present between the actual vs predicted values to validate the quality of the surrogate model (and hence the relevance of the probabilistic analysis) is based on CFD alone? It's a bit strange that you have the measured data available, but don't use it to comment on the quality of the surrogates. Is the CFD good enough to predict the local performance deterioration? Given the interest of probabilistic strategies to assess in-service wear and erosion, I find this a shortcoming for a journal paper.

2) You state that input variables are typically correlated for turbomachinery applications. This is partially true during a design phase where you radially interpolate between sections and/or use Bezier-parametrization or FFD approaches to generate smooth surfaces. It is also partially not true (no link between LE angle and TE angle between two distinct blade sections) so I would not use it as a general remark. In addition, and especially in the context of probabilistic analyses to assess erosion and wear, that correlation will reduce/vanish due to local impacts. Can you really show, based on the measured blades, that these input variables are correlated? And related to my first remark, can you confirm that the impact of averaging the domain is still valid? Is the reduction in dimensionality combined with the CFD good enough to be relevant for probabilistic analyses on such geometry deformations?

Comments on the Quality of English Language

There is a regular misuse of commas that are introduced without any reason and that feels like a general "find and replace action" that has not been proofread afterwards. Please correct.

Reviewer 2 Report

Comments and Suggestions for Authors

1. According to the title "Improvements in Probabilistic Strategies ..", I would expect that the authors suggest improvements to the existing methods. The authors review existing methods. I suggest to change the title.

2. The abstract does not summarizes sufficient the scope of this work. Reading the abstract first, the reader cannot understand easy what is exactly the scope of this work. 

3. Introduction: rewrite the first paragraph and provide better description of the problem. You mention "examples are the uncertainty quantification and sensitivity analysis" but be more specific - how does this apply in turbomachinery applications?

4. Introduction - line 25 "high surrogate model": what do you mean by "high"?.

5. "For most methods, uncorrelated input variables are required": Sensitivity analysis does not necessarily require that input variables be uncorrelated. In fact, part of the value of conducting a sensitivity analysis can be to understand how dependencies or correlations between input variables affect the output of a model. Can you please analyze this more and add references?

6. Rewrite the Introduction - it is not well-written. Also include better description of the problem, the current state-of-the-art and the limitations. Add more references. Also include a paragraph that describes what is the specific scope of this work and the novelty of this work? 

7. Line 51 - 52 "Probabilistic methods  can be used to quantify this impact." --> This sentence is not correct. You want to quantify the uncertainty of the performance parameters. 

8. Section 2.1 - 2nd paragraph: Improve the description. Write it in a better sequence. 

9. Line 64 - 66 "For  the sample size, a compromise must be found between accuracy (large sample size) and  numerical effort (low sample size)": Since at this point you refer at this challenge which is an important  issue at the design of experiments, you do not mention anything about active sampling. Active sampling is a strategy used in machine learning where the learning algorithm selectively queries or chooses the data points from which it learns. The main goal of active sampling is to improve model performance with fewer training samples by intelligently selecting the most informative examples, rather than randomly choosing from a pool of data. You can check this paper: Zhang, J., Cammarata, L., Squires, C., Sapsis, T. P., & Uhler, C. (2023). Active learning for optimal intervention design in causal models. Nature Machine Intelligence5(10), 1066-1075

10. Section 2.3.1: You don't need to mention all these formulas (10) - (16). Just keep formula (9) and provide a brief explanation of R2 - why this metric is usually used?

11. Section 2.3.2, statement in the first paragraph (lines 132-134): The idea is to examine the R2 also in test data. Please rewrite this section and make it more coherent.

I assume that this is the message you want to convey "While R2 can be used to evaluate test data, the quality of this evaluation can vary significantly depending on how the test data is chosen. A single split of the data into training and test sets might not provide a complete picture of the model's performance due to the randomness in the split or potential biases in the data. In contrast, R2cv obtained through cross-validation offers a more balanced and comprehensive evaluation, as it averages the model's performance over multiple splits, thus providing a better measure of the model's generalization ability."

However, do you used only two subsets for the k-fold cross validation? This is not enough.

Also why don't you calculate the R2 on training and test data?

12. Section 3.1: The sampling is not necessary to be uniform. This depends on the application and what the surrogate model should predict. For instance check Figure 4.4 in Blanchard, A., & Sapsis, T. (2021). Output-weighted optimal sampling for Bayesian experimental design and uncertainty quantification. SIAM/ASA Journal on Uncertainty Quantification9(2), 564-592.

13. Section 4.1: Can you provide an illustrative example based on application of turbomachinery instead of the restaurant order?

14. A general comment is that I do not see the innovation/novelty in this work. Please highlight what is the novelty in the Abstract, Introduction and Summary.

Comments on the Quality of English Language

Minor improvements are suggested.

e.g. Line 48 "which is subject" --> I guess you want to say: which are subject

e.g. Line 85 "An alternative are" --> An alternative  is

Reviewer 3 Report

Comments and Suggestions for Authors

Report of Review

 This paper discusses the sampling methods used in the optimization of turbomachinery and analyzes their sensitivity.  

Main remarks

1. This article must be clearly stated “Review article” not a “Research article”.

2. Check thoroughly all the mathematical relations.

3. Please provide citation where it is necessary. Some information is used from the literature without any reference. The whole section 3 is an example and there are many other sections.

4. All the test cases should be presented beforehand and discussed; some are just dropped in figures.

5. Comparisons made in Figure 5 are based on what?

6. Figures and tables must be shown after their introduction or discussion parts.

7. What you mean here : test case a 15D mathematical

8. Check the mathematical symbols, such as used in equation (19). Is x of vector?

9. The example presented in table 1 unclear and the consequence analysis in  4.1.1. Sobol Indices

10. The symbol dollar represents a cost?

11. Figure 10. of bad quality both for plane view and scales.

12. What profile is this NACA-like

13. In section 5.1. Turbomachinery Test Case: aerodynamic improvements should be presented and tabulated and discussed.

Queries

14. The language need to be checked carefully, there are many typos, and grammatical errors.

15. The abstract should present the real contribution in view of similar works. What problem was studied and why is it important? What is the novelty of the work? What methods were used? What are the important results?

16. In abstract what you mean by “coefficient of importance

17. Add a list of abbreviation for acronyms

18. Correct here : Another issuer

19. Check citation style in the text

20. Change here : For each realization

21. Correct here : when in input

22. Explain k-fold

23. Correct here : More details can are given by

24. What this acronym refer to : oLHS is this a modified LHS

25. What this acronym refer to : Elhs

26. What case presented in Table 1.

27. NAVIER-Stokes should be capital

28. Not clear here :clear outliers can be seen

29. Conclusion has long introductory paragraphs but not pertinent outcomes, recommendations and perspectives.

Comments on the Quality of English Language

minor corrections

Round 2

Reviewer 2 Report

Comments and Suggestions for Authors

1. Please address the comment #1 from my previous review.

2. Introduction must be improved. The authors claimed lack of time, but this does not correspond as an answer to the comment. 

3. The novelty of this work remains unclear. 

Comments on the Quality of English Language

No comments. 

Reviewer 3 Report

Comments and Suggestions for Authors

1. Complete the list of acronyms: example:    Elhs

2. Not clear here: outliers  change this word

3. What profile has been used is this an NACA-like if not give its nomenclature

4. In section 5.1. Turbomachinery Test Case: aerodynamic improvements should be presented tabulated and discussed.

5. Figure 10. Flow Field around Blade ( not clear what you are presenting in the same scales used and the 3-d view does not well describe the flow details. please check with some CFD results used in compressor blades and get inspired about how to show the results

Round 3

Reviewer 2 Report

Comments and Suggestions for Authors

No more comments

Reviewer 3 Report

Comments and Suggestions for Authors

Accept

Comments on the Quality of English Language

Accept